# A New Multi-input Model with the Attention Mechanism for Text Classification

## Abstract

Recently, deep learning has made extraordinary achievements in text classification. However, most of present models, especially convolutional neural network (CNN), do not extract long-range associations, global representations, and hierarchical features well due to their relatively shallow and simple structures. This causes a negative effect on text classification. Moreover, we find that there are many express methods of texts. It is appropriate to design the multi-input model to improve the classification effect. But most of models of text classification only use words or characters and do not use the multi-input model. Inspired by the above points and Densenet (Huang et al. (2017)), we propose a new text classification model, which uses words, characters, and labels as input. The model, which is a deep CNN with a novel attention mechanism, can effectively leverage the input information and solve the above issues of the shallow model. We conduct experiments on six large text classification datasets. Our model achieves the state of the art results on all datasets compared to multiple baseline models.

## 1 Introduction

Text classification, including sentiment analysis (Pang et al. (2002); Yang & Cardie (2014)), topic classification (Tong & Koller (2002)), and spam detection (Jindal & Liu (2007)), is an important subdomain of natural language processing (NLP). It is widely used in public opinion monitoring, advertising filtering, user and product analysis and other fields. In recent years, deep learning has been frequently used in text classification. Compared with traditional methods (Wang & Manning (2012)) that rely on hand-crafted features, deep learning methods are employed to learn features from texts by a variety of neural network structures, especially recurrent neural network (RNN) and convolutional neural network (CNN), and attention mechanisms. There are some examples of text classification in Table 1. Both the first and the second cases should be classified as the topic of science and technology, and the third case should be classified as the topic of animals. Most of key words or phrases of case1 and case2, which determine the category, are similar (e.g., NASA, moons, and Saturn), but the location of them is different.It can be well captured by CNN, because of the position invariance of CNN. However, the key words or phrases of case3 are scattered. The dependencies between them are long distance. Compared with case1 and case2, these key words or phrases are relatively hard to capture by straightforward or shallow models. And case3 contains a lot of geographical names, which make it possible to be misclassified as geographic topics.

By the above examples, we find a problems in present models: These models are shallow, especially CNN. It limits the final classification effect. Specifically, texts have the hierarchical structure which includes characters, words, phrase, sentences, and documents. The shallow models are hard to capture the hierarchical, high-level, long distance or global features form texts, which is why the models are limited.. Nowadays, the amount of text information on the Internet is increasing rapidly. Thus, the model which can cope with texts of various lengths and levels is needed, especially deep models of complicated texts. Compared with the CNN models (He et al. (2016); Huang et al. (2017)) of computer vision (CV), the CNN models of text classification are very shallow. It indicates that deeper models can be tried to improve the result of text classification, as they have done in CV.

Moreover, most of models utilize words or character as the single input, but the multi-input model is well suited to text classification. The natural language has a variety of carriers or expressions, such as words and characters, which can be related to others. This is very helpful for designing multiple

Table 1: Examples of text classification. The Bold is the key word or phrase.

| | |
|---|---|
| **case1**: | Two new moons were spotted around **Saturn** by the Cassini **space probe**, raising the total to 33 **moons** for the ringed **planet**, **NASA** said on Monday |
| **case2**: | **NASA**'s Cassini **spacecraft** has spied two new little **moons** around satellite-rich **Saturn**, the space agency said Monday. |
| **case3**: | Rendall's serotine (Neoromicia rendalli) is a species of vesper **bat**.It is found in Benin Botswana Burkina Faso Cameroon Central African Republic Chad Republic of the Congo Democratic Republic of the Congo Gambia Ghana Kenya Malawi Mali Mozambique Niger Nigeria Rwanda Senegal Sierra Leone Somalia South Africa Sudan Tanzania Uganda and Zambia.Its **natural habitats** are dry savanna moist savanna subtropical or tropical dry shrubland and subtropical or tropical moist shrubland.It is threatened by **habitat** loss. |

inputs. We think the multi-input model can obtain better results because it provides more useful information. Some related works (Amplayo et al. (2018); Xue & Li (2018)) have been done in order to solve this problem, but most of works used multi-input models in some special subdomains such as user sentiment analysis. Those are not the general methods for text classification.

In order to solve these problems, we propose a novel deep multi-input model for text classification. Our model uses words, characters, and labels as inputs. These are proved to be beneficial for text classification in previous works. Both the multi-input model and deep model have relatively more parameters and more expensive computing cost, and more effective feature extraction is needed on the multi-input model. Thus, we use a structure that is similar to Dense block of Densenet (Huang et al. (2017)) to alleviate the above problems, because it has narrow feature maps and can reduce the numbers of parameters. Moreover, we employ a new attention mechanism, which allows capturing of dependencies without regard to their distance in inputs, to replace the global pooling. It can strengthen feature extraction from several inputs. We conduct extensive experiments on several public text classification datasets. The results show that our model outperforms all baseline models on all datasets, and has fewer parameters in comparison to similar works.

Our primary contributions are as follows:

1. We propose a novel multi-input model for text classification. Our model inputs include words, characters, and labels. Most of previous works only use a single input such as the word or use the multi-input model in certain subdomains like user sentiment analysis (Amplayo et al. (2018)).

2. Inspired by Densenet, we design a 42-layers network. It is deeper than most of present models. Compared with shallow and straightforward models, our model can extract more global or long-term features, and have fewer parameters.

3. We use a new attention mechanism in our model. It is used to replace the global pooling that commonly used in CNN. Compared with global pooling, our method can effectively utilize inputs information to capture key part more precisely. Compared with using the attention mechanism at the beginning of the model, this method can significantly reduce parameters and computation, because the feature maps of high layers are much smaller than that after embedding.

## 2 RELATED WORK

Deep learning has been shown their values in many domains of artificial intelligence, including text classification. This paper will introduce some significant works about it in text classification.

For the past few years, deep learning or deep neural networks have been considered the most prevalent machine learning method in the field of text classification. Most of them use pre-trained word embedding methods such as word2vec (Mikolov et al. (2013)) or glove (Pennington et al. (2014)) and models based on CNN or RNN, but some of them employ the word embedding without pre-trained (Joulin et al. (2017)) or other embedding methods such as utilizing characters (Zhang et al. (2015)).

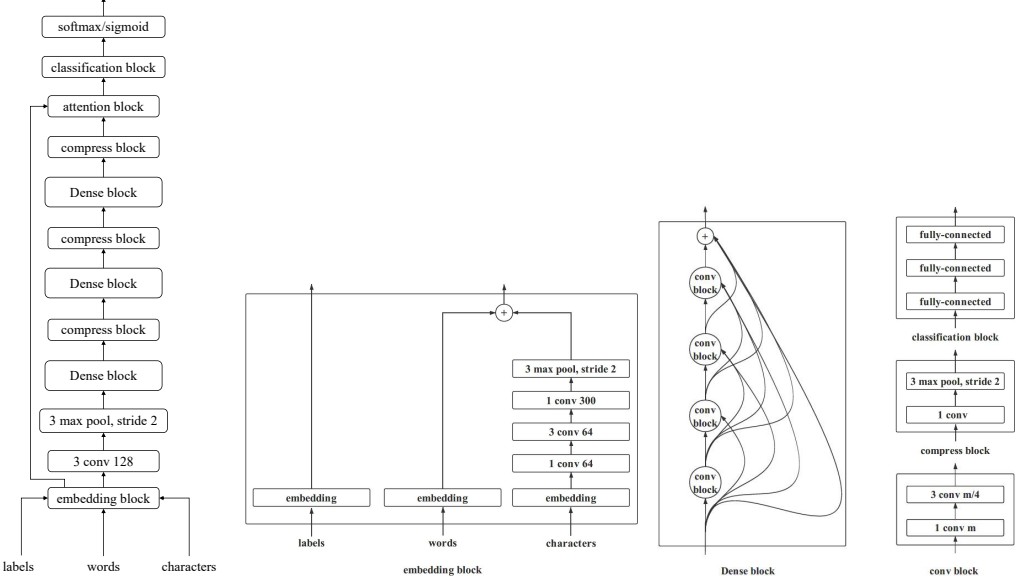

Figure 1: Our model architecture.

CNN is regarded as the predominant model in the computer vision due to its friendliness in obtaining local features and processing in parallel. Recently it also has achieved high performance in NLP. In text classification, Kim (2014) combines several 1D CNNs to classify sentence. The convolution filter widths and feature maps are various between these 1D CNNs. Similarly, Kim (2014) also use convolutional layers with several kinds convolution filter widths. And they propose the dynamic $k$-max pooling, the $k$ of which is related to the sentence length and the network depth. Zhang et al. (2015) propose a character-level CNN model. The model consists of 6 convolutional layers and uses English letters, digits and punctuations instead of the word to represent text. Their work shows that characters can be used as the input of text classification and the characters method can obtain good results. Zhang et al. (2017) propose a representation method which use the convolutional encoder and deconvolutional decoder to classify text. The convolutional encoder has the stride structure and no more than 4 layers. Xue & Li (2018) propose a CNN model with a gating mechanism to figure out the tasks of aspect based sentiment analysis and achieve excellent performance. It is enlightened from recent advances in computer vision (He et al. (2016)), Conneau et al. (2017) design a 29-layer deep CNN model to improve the performance on large datasets.

The NLP input and the sequence input are highly compatible. Therefore, RNN variants, such as long short term memory (LSTM) (Hochreiter & Schmidhuber (1997)), are often used in NLP tasks due to the suitable for sequence features. Tang et al. (2015a) propose the gated recurrent neural network, which employs LSTM and gated neural network to solve the document sentiment classification task. The attention mechanisms, especially Transformer (Vaswani et al. (2017)), have achieved great results in many fields of NLP. Most attention mechanisms are used with RNN to extract features more accurately. Zhou et al. (2016) propose an attention mechanism based on bidirectional LSTM in cross-lingual sentiment classification. Yang et al. (2016) design a hierarchical attention network (HAN) that can better capture document structure features. It uses bidirectional GRU and attention mechanism on words and sentences, respectively. Moreover, Lai et al. (2015) propose the recurrent convolutional neural networks (RCNN), which add a recurrent structure before convolutional layers. It can utilize the left and right contents of the word to improve the performance of classification.

Moreover, labels, as the potential information, can effectively extend the input to help improve the precision of the model (Yogatama et al. (2015)). However, it is not employed much in text classification. Tang et al. (2015b) propose a heterogeneous text network, which utilizes an embedding including the labeled information and different levels of word co-occurrence information. This embedding has a good performance for the particular task. Zhang et al. (2018) propose a multitask label-embedding to transform labels into semantic vectors, which utilizes semantic correlations among tasks to enhance performances.

## 3 METHODS

Our model is briefly illustrated in Figure 1. It mainly consists of embedding block, Dense block, compression block, attention block and classification block. The embedding block, including words, characters, and labels, is the first part of this model. There are a convolutional layer and a pooling layer after the embedding block. And then the combination of Dense block and compression block is repeated several times, but it is an attention block after the final Dense block instead of a compression block. The classification block is the final part. It consists of three full-connected layers, followed by dropout. The max pooling is used for all pooling layers, and batch normalization (BN) (Szegedy et al. (2015)) is applied after all convolutional layers. The activation function is the rectified linear unit (RELU) for all convolutional layers and full-connected layers.

**Embedding block**  The word embedding and the label embedding are similar to common pre-trained word embedding, but we use three consecutive operations after the word embedding layer: dropout, adding Gaussian noise and BN. Inspired by character-level CNN (Zhang et al. (2015)), the characters embedding utilizes multi-layer CNN to encode the character information. The CNN encoder has a max pooling layer in the last. And to reduce the parameters, its structure refers to the Inception (Szegedy et al. (2015)). We also use three identical consecutive operations as the word-level embedding after the character-level embedding layer. Finally, we concatenate the words embedding part and the characters embedding part as the first output, which is used as the input of the convolutional layer after the embedding block. The labels embedding part is the second output, which is used as one of the inputs of attention block. Comparing with using the word embedding only or the character embedding only, our methods can express text more fully and extract more detailed information, which can help to improve the model performance. Figure 1 shows the embedding block architecture.

**Dense block**  The structural design of this block is based on Densenet (Huang et al. (2017)), so we call it Dense block. Dense block consists of several convolutional blocks, which have two $1D$ convolutional layers. The filter widths of the first convolution layer and the second are 1 and 3, and the size of feature maps of the first layer is 4 times that of the second layer. It can reduce the number of feature maps, and thus to make computational cheaper. Each convolution block is directly connected to all the other convolutional blocks. In a Dense block which has $X$ convolutional blocks, there are $X(X-1)/2$ connections between convolutional blocks. However, in $X$ layers typical CNN, there are only $X$ connections. In other words, the final output includes all previous convolutional blocks output and the initial input, not just the output of the last convolutional layer as the typical CNN model. In this block, features are utilized efficiently to reduce relearning, and the change of gradient from loss function can affect each layer more directly to address the problem of vanishing-gradient. It is substantially beneficial for designing very deep models. Thus, our model can be deeper and perform better than the typical CNN model. Figure 1 shows the Dense block architecture.

Noticeably, the input size of the convolutional block will grow gradually, because each convolution block input includes the output of all the previous convolution blocks and the initial input. If $m$ is the size of feature maps of the second layer of each convolution block, and that of the first should be $4m$, the input feature maps of the next convolutional block will increase by $m$ compared to the previous. Define $M$ as the size of input feature maps of the $h$-th convolutional block:

$$M = M_0 + g \times (h-1) \tag{1}$$
$$g = i \times n \tag{2}$$

Where $M_0$ is the size of input feature maps of the first convolutional block, $g$ are the number of growth feature maps, $i$ is the number of channels in the growth feature maps, $n$ is the dimensions of feature maps. We call the hyperparameter $i$ the growth rate of the model. It can control the amount of information that each convolutional block gives the final output. The growth rate is usually small but enough to obtain an excellent result. Therefore, the feature maps of convolutional blocks are very narrow. It makes our model have fewer parameters. Since Dense block is used repeatedly, our model become very deep and can better extract long-range associations, global representations, and hierarchical features. It is why this model is called the multiple shortcut connections CNN.

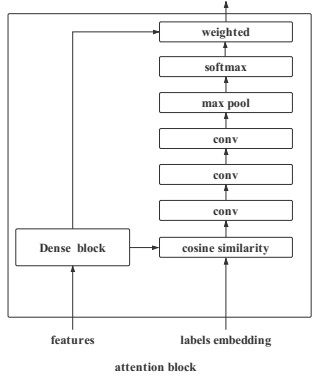

Figure 2: The attention block architecture.

**Compression block**   The feature maps of our model, especially the number of channels, are increased due to the Dense block. If the Dense block is repeated several times, the feature maps may become huge. After the Dense block, we design a structure which is used to compress the feature maps to avoid this problem. The 1D max pooling though can perform downsampling to reduce the number of feature maps, but the number of channels is usually not changed. Thus, a $1D$ convolutional layer with 1 convolution filter width is used before the max pooling layer. The number of channels of this convolutional layer is compressed to that of the previous Dense block input. We call this combination of convolutional layer and max pooling layer as the compression block. After the compression block, the feature maps have been compressed on both dimensions and channels. In other words, this block makes feature maps not only not become huge and harmful, but also reduce periodically to improve the model efficiency. Figure 1 shows the compress block and the classification block architecture.

**Attention block**   The traditional CNN methods for text classification usually use global pooling or $k$-max pooling after the last convolutional layer. The attention block is different from conventional pooling operations, which is roughly filtered out one or several features. It utilizes the relation between labels and extracted features to give suitable weights to extracted features. Most labels text usually short and recapitulative, which are much simpler than common texts. Thus, extracting label features do not need complex structures such as that of common texts to increase computational complexity. We choose the label embedding and the features extracting from texts and characters as the block input. It has smaller and narrower feature maps in comparison to designing an attention mechanism after the embedding block, so it has fewer parameters and avoids using more complex attention structures to deal with huge feature maps. Figure 2 shows the attention block architecture.

The input of this block are the labels embedding and the output of the last compression block. All labels are concatenated as a sentence. In the embedding block, we embed the label sentence as well as the word-level embedding. Then we use average pooling to map each label to a vector. Besides, we employ one Dense block to extract global text features further and convert the features to the dimension of label-embedding. Define $\boldsymbol{L} = [\boldsymbol{L}_{:,1}, \boldsymbol{L}_{:,2}, \ldots, \boldsymbol{L}_{:,k}]$ as the label-embedding, $\boldsymbol{V} = [\boldsymbol{V}_{:,1}, \boldsymbol{V}_{:,2}, \ldots, \boldsymbol{V}_{:,d}]$ as the extracted features of a sample, which are gotten from the features extracting step. Where $k$ is the number of classes, $d$ is the dimension of the extracted feature maps. We use the cosine similarity $\boldsymbol{C} \in \mathbb{R}^{d \times k}$ to describe the degree of correlation between $\boldsymbol{L}$ and $\boldsymbol{V}$.

$$
\boldsymbol{C} = \begin{pmatrix} C_{1,1} & & & & \\ & \ddots & & & \\ & & C_{i,j} & & \\ & & & \ddots & \\ & & & & C_{d,k} \end{pmatrix} \tag{3}
$$

Where $C_{i,j} = \frac{\boldsymbol{V}_{:,i}^{T} \cdot \boldsymbol{L}_{:,j}}{|\boldsymbol{V}_{:,i}^{T}| \cdot |\boldsymbol{L}_{:,j}|}$ is the cosine similarity between the $i$-th feature vector $\boldsymbol{V}_{:,i}$ and $j$-th label-embedding vector $\boldsymbol{L}_{:,j}$. The cosine similarity is a simple way that is difficult to obtain relative information between labels and extracted features fully. To generate more precise attention weights,

Table 2: Summary statistics of datasets.

| Datasets | Classes | Train Samples | Test Samples | Average Words | Word Lengths | Character Lengths | Batch Sizes |
|---|---|---|---|---|---|---|---|
| AG | 4 | 120,000 | 7,600 | 45 | 50 | 300 | 512 |
| Sogou | 5 | 450,000 | 60,000 | 578 | 1,300 | 5,000 | 32 |
| Yahoo | 10 | 1400,000 | 60,000 | 112 | 223 | 1358 | 128 |
| Dbpedia | 14 | 560,000 | 70,000 | 55 | 100 | 610 | 256 |
| Yelp.p | 2 | 560,000 | 38,000 | 153 | 337 | 1350 | 128 |
| Yelp.f | 5 | 650,000 | 50,000 | 155 | 305 | 1194 | 128 |

Table 3: Test error rates (%) on six datasets. The Bold is the best result.

| Methods | AG | Dbpedia | Yelp.p | Yelp.f | Yahoo | Sogou |
|---|---|---|---|---|---|---|
| Linear model [30] | 7.64 | 1.31 | 4.36 | 40.14 | 28.96 | 2.81 |
| FastText [8] | 7.5 | 1.4 | 4.3 | 36.1 | 27.7 | 3.2 |
| Region.emb [16] | 7.2 | 1.1 | 4.7 | 35.1 | 26.3 | 2.4 |
| LSTM [30]) | 13.94 | 1.45 | 5.26 | 41.83 | 29.16 | 4.82 |
| char-CNN [30] | 9.51 | 1.55 | 4.88 | 37.95 | 28.80 | 4.88 |
| word-CNN [30] | 8.55 | 1.37 | 4.60 | 39.58 | 28.84 | 4.39 |
| D-LSTM [28] | 7.9 | 1.3 | 7.4 | 40.4 | 26.3 | 5.1 |
| VDCNN [2] | 8.67 | 1.29 | 4.28 | 35.28 | 26.57 | 3.18 |
| char-CRNN [23] | 8.64 | 1.43 | 5.51 | 38.18 | 28.26 | 4.80 |
| Our model (word) | 7.17 | 1.10 | 3.95 | 34.85 | 26.22 | 2.40 |
| Our model (word+character) | **6.53** | **1.09** | **3.34** | **34.4** | **25.89** | **2.36** |

we utilize the CNN structure to capture further location correlation. We use a multi-layer CNN structure with small convolution filter width. It results in fewer parameters and larger coverage of the convolution filter. Therefore, this block has an advantage when facing with large feature maps of complex models. After multi-layer CNN structures, the max pooling is employed to select the most relevant vector $\boldsymbol{h} = [h_1, h_2, \ldots, h_d]$ between labels and extracted features. We can compute the $i$-th attention score/weight $a_i$ by $\boldsymbol{h}$:

$$a_i = \frac{exp(h_i)}{\sum_{j=1}^{d} exp(h_j)} \tag{4}$$

Where $h_i$ is the $i$-th element of $\boldsymbol{h}$. The final text representation $\boldsymbol{t}$ is the sum of weighted features:

$$\boldsymbol{t} = \sum_{i=1}^{d} a_i \boldsymbol{V}_{:,i} \tag{5}$$

The final output $\boldsymbol{t}$ is used as the input of the classification block. The attention block can assign weights between labels information and features extracting by words and characters more accurately. In other words, it can take full advantage of a large amount of information of the multi-input model. Thus, our multi-input model with this label-attention mechanism is capable of improve classification accuracy effectively.

## 4 EXPERIMENTS

### 4.1 DATASETS

We experiment on six public datasets[1], which are proposed by Zhang et al. (2015). Table 2 shows the summary statistics. AG and Sogou are news topic classification datasets. Dbpedia is an ontology classification dataset. Yahoo is a topic classification dataset about Q&A. Yelp.p and Yelp.f are review sentiment classification datasets, the former is coarse granularity (two classes: positive and negative), and the latter is fine granularity (five classes). Sogou is the Chinese pinyin dataset, the others are English text datasets.

### 4.2 THE EXPERIMENT SETUP

We randomly selected 10000 documents from the training data as the validation set. In word embedding part, we tokenized all the English texts with NLTK (Bird & Loper (2004)) and used the 300d

---

[1]https://github.com/zhangxiangxiao/Crepe

Table 4: Comparison of model parameters

| Methods | CNN | LSTM | VDCNN | | | | Our model |
|---|---|---|---|---|---|---|---|
| | | | 9-layers | 17-layers | 29-layers | 49-layers | |
| Parameters | 541 k | 1.8 M | 2.2 M | 4.3 M | 4.6 M | 7.8 M | 1.1-1.2 M |

Table 5: Test error rates (%) for different variations of our model. We run all experiments on AG and Yelp.p.

| Methods | Embedding | Dense block | Convolution Block | Growth Rate | AG | Yelp.p |
|---|---|---|---|---|---|---|
| Our model(global pooling) | word | 3 | | | 8.67 | 5.35 |
| | word+character | 3 | 4 | 32 | 7.26 | 4.51 |
| Our model (single) | | 4 | | | 6.72 | 3.57 |
| | word | 4 | | | 7.17 | 3.95 |
| | | 2 | 4 | 32 | 7.36 | 3.72 |
| | | 3 | | | 6.86 | 3.56 |
| | | 4 | | | **6.53** | **3.34** |
| | | 5 | | | 6.96 | 3.70 |
| Our model | | 6 | | | 6.92 | 3.82 |
| | word+character | | 2 | | 7.19 | 3.96 |
| | | 4 | 6 | 32 | 6.84 | 4.04 |
| | | | 8 | | 6.55 | 3.72 |
| | | | | 8 | 6.89 | 4.52 |
| | | 4 | 4 | 16 | 7.03 | 4.70 |
| | | | | 48 | 7.03 | 3.57 |
| | | | | 64 | 6.72 | 3.85 |

glove 840B vectors by Pennington et al. (2014) as initialization. In addition, we used the word2vec (Mikolov et al. (2013)) to train the pinyin representation, which is used as initialization of the Sogou dataset. In character embedding part, the processing was like Zhang et al. (2015). The characters consist of letters (all uppercase letters were converted to lowercase), digits and punctuations. We initialized the character embedding weights using a Gaussian distribution. The initial mean and standard deviation were (0,0.05). Word and character lengths were set according to the datasets. They are also shown in Table 2. The drop rate of both parts was set to 0.5. Empirically, A convolutional layer of 128 convolutions of size 3 was applied to the first after embedding block. We used 4 Dense blocks in our model, each including 4 convolution blocks. And the growth rate was set to 32. The max pooling with size 3 and stride 2 was used for all pooling layers except to the global max pooling. The feature maps were halved in both dimension and channel after each compression blocks. We performed three full-connected layers with size 256 and dropout rate 0.2 or 0.5 in the classification block. Moreover, in the attention block, the processing was similar to the word embedding part. We employed 3 convolutional layers with filter width 3 or 4, the number of channels of which was the same as the number of the classes. Finally, we selected Nadam as the optimizer. The initial learning rate was set to 0.002. We used cross entropy as the loss function. All experiments were performed on the NVidia 1080Ti GPU, using different mini-batch sizes on different datasets as shown in Table 2.

## 4.3 BASELINES

We compared our method using with various baseline methods, including CNN, RNN, the hybrid method, and other methods. The CNN based methods consist of the word-CNN, char-CNN [30] and the 29-layers very deep CNN (VDCNN) (Conneau et al. (2017)). The RNN based methods consist of LSTM (Zhou et al. (2016)) and the discriminative LSTM (D-LSTM) (Yogatama et al. (2017)). The char-CRNN (Xiao & Cho (2016)) is the hybrid method based on the combination of CNN and RNN. The other methods include the linear model (Zhang et al. (2015)), the FastText (Joulin et al. (2017)), and a region embedding method (Qiao et al.). We also tested our model without characters inputs.

## 4.4 RESULTS

**Error rates** Table 3 shows the test error rates results. The simple methods (rows 1-3) perform better on small datasets such as AG than the large such as Yelp.f. The sophisticated methods (rows 7-9) are the opposite. The result is consistent with common sense that the demand degree of the number of training data is usually positively correlated with model complexity. Simple models may even have an advantage on small datasets. The best simple baseline is region embedding method (Qiao et al.), and the best complex baseline is VDCNN (Conneau et al. (2017)). Compared with

them, our methods including 2 embedding structures variants work well on both small and large data sets. It ranks the 1st (word+character) or 2nd (word) on all datasets. The results of long lengths datasets validates that our model can extract more global or long-term features. And the result of our model with the combination embedding of words and characters outperforms that of only word embedding, which validates the effectiveness of the characters input. We observe error reduction are less significant on Yahoo, Sogou, and Dbpedia. There may be some reasons: ($i$) The training samples of Sogou and Dbpedia are relatively small, and Yahoo have a special text structure (Q&A). Both may require more data to training for complex methods because all the complex baselines do not perform well on those datasets. ($ii$) The number of classes of Dbpedia and that of Yahoo are fourteen and ten respectively, and the others are no more than five. It is relatively more difficult to train label-attention mechanism to get the most similar labels exactly when there are relatively more classes. ($iii$) Sogou is a Chinese pinyin dataset. Unlike other datasets, the pre-train pinyin word vector is got by Sogou dataset. Compared with 300d glove 840B vectors (Pennington et al. (2014)), the number of pre-train samples of pinyin is much smaller. Thus, the representation of words and labels may become less effective on Sogou dataset.

**Parameters** We compared the parameters with CNN, LSTM, and VDCNN. Table 4 shows the parameters results. Except for shallow CNN with very few parameters, our model has far fewer parameters than LSTM and the shallowest VDCNN which have the least number of parameters. And our model (42-layers) is deeper than all the baseline except for 49-layers VDCNN. It means that our method avoids excessive parameters while achieving a low error rate.

**Impact of structures** We compared the impact of using different structures on our model as shown in Table 5, including embedding structures, attention block structures, and other hyperparameters. We tested 2 embedding structures variants (using global pooling and attention block): only word embedding and the combination embedding of words and characters. We also tested two attention block structures: a single convolutional layer with large convolution filter width (55) and three convolutional layers with small convolution filter width (3). Besides, we compared the impacts of different growth rates, the number of multiple direct connections blocks, and the number of convolution blocks on the results. The results show that the combination embedding is better than only word embedding (rows 1 and 2, 4 and 7), our model with attention block outperformed that without attention block (rows 1 and 4, 2 and 3, 2 and 7), three convolutional layers attention block is better than the single convolutional layer (rows 3 and 7), and the best combination is 4 Dense block with 4 convolution blocks and growth rate $i = 32$ (rows 5 -16).

## 5 CONCLUSION AND FUTURE WORK

This paper proposes a novel deep CNN model with a new label-embedding attention mechanism for text classification. Our model adds shortcut connections between any two convolutional layers. Compared to the traditional deep model and attention mechanism, our method has fewer parameters, deeper structure, and takes full advantage of the inputs information. Therefore, our model achieves the state of the art results on several public text classification datasets. Moreover, we also analyze the effects of different structures on the method and the reasons for the relatively poor performance of the method. In the future, we will apply and improve our method to the multi-label problem and the classification of obscure label information.

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
