# OpenReview forum: "A New Multi-input Model with the Attention Mechanism for Text Classification"
_ICLR.cc/2020/Conference — Reject_

### Official Review · AnonReviewer3 · 2019-10-22
**Official Blind Review #3**

**Rating:** 1

**Review:**

Summary

This paper introduces a new model architecture for doing text classification. The main contribution is proposing a deeper CNN approach, using both word and character embeddings (as well as label embeddings with attention). The paper claims improved performance over baselines.

Decision

I reject the paper for 3 main reasons:
1) Very misleading claims regarding establishing a new state of the art. The baselines used for comparison don't include any of the best existing published results.
2) Lack of positioning within the literature. In particular, no mention nor discussion of Transformers (self-attention) networks, including BERT and XLNet approaches, which are the state of the art in text classification.
3) Lack of justification/explanation for the proposed architecture. One key argument made is that current models are shallow, but it appears that only CNN models are considered for that comparison. More discussion is needed to understand why the new aspects of the proposed network are importantly different from other existing approaches.

Additional details for decision

The results from this paper are significantly inferior to the best results published. With a few quick searches, I found that there are several approaches performing better than the proposed model on every dataset considered in the analysis, as you can see below.

http://nlpprogress.com/english/text_classification.html
https://github.com/sebastianruder/NLP-progress/blob/master/english/sentiment_analysis.md
https://paperswithcode.com/sota/text-classification-on-yahoo-answers

Extra notes (not factoring in decision)

- Consider spacing out the 3 rightmost blocks in Figure 1, I found the layout confusing and there's space available.
- In section 3, I would have liked more explanation for the motivation of the various design choices.

**Experience Assessment:**

I have read many papers in this area.

**Review Assessment: Checking Correctness Of Derivations And Theory:**

N/A

**Review Assessment: Checking Correctness Of Experiments:**

I carefully checked the experiments.

**Review Assessment: Thoroughness In Paper Reading:**

I read the paper at least twice and used my best judgement in assessing the paper.

---

### Official Review · AnonReviewer1 · 2019-10-24
**Official Blind Review #1**

**Rating:** 1

**Review:**

This paper presents a multi-input model for text classification.  The presentation of this paper is very poor.  The novelty is very limited.  The justification of the proposed architecture is not persuasive.  The experiment design has many flaws.  All the compared baselines are extremely weak.  The authors need to follow the experiment setup in more recent text classification papers.  Authors need at least to compare their methods to BERT.  The current version of this paper is definitely not an ICLR publication.

**Experience Assessment:**

I have published in this field for several years.

**Review Assessment: Checking Correctness Of Derivations And Theory:**

N/A

**Review Assessment: Checking Correctness Of Experiments:**

I assessed the sensibility of the experiments.

**Review Assessment: Thoroughness In Paper Reading:**

I made a quick assessment of this paper.

---

### Official Review · AnonReviewer4 · 2019-11-04
**Official Blind Review #4**

**Rating:** 1

**Review:**

This paper described a multi-input model for text classification. This paper is a bit hard to read compared to other submissions. I highly recommend authors to have native speaker to proofread this before submission. The paper mentioned "labels" as input to embedding layer as well as attention mechanism. However, it will be more helpful if authors can provide examples of labels. Moreover, the experiment results (e.g., Table 3 and 5) did not include label. I wonder if the methods used as baselines are still the state-of-the-art as language model pre-training might have better results. It will be more convincing if authors include additional baselines such as fine-tuning BERT.

Minor issue: the references in Table 3 (in brackets) are hard to find.

Based on above reason, I would reject this paper.

**Experience Assessment:**

I have published one or two papers in this area.

**Review Assessment: Checking Correctness Of Derivations And Theory:**

I did not assess the derivations or theory.

**Review Assessment: Checking Correctness Of Experiments:**

I assessed the sensibility of the experiments.

**Review Assessment: Thoroughness In Paper Reading:**

I made a quick assessment of this paper.

---

### Decision · Program_Chairs · 2019-12-19

**Decision:**

Reject

**Comment:**

This paper proposes a CNN-based text classification model that uses words, characters, and labels as its input. It also presents an attention block to replace the pooling operation that is typically used in a CNN. The proposed method is evaluated on six benchmark classification datasets, achieving reasonably good results.

While the proposed method performs reasonably well compared to baselines in the papers, all reviewers pointed out that there is no discussion or comparison with existing SotA based on pretrained models (e.g., BERT, XLNet), which would strengthen the main claim of the paper. All three reviewers also suggested that the writing of the paper could be improved. The authors did not respond to these reviews, so there was little discussion needed to arrive at a consensus.

I agree with all reviewers and recommend to reject the paper.